# Impact of COVID-19 Vaccination on Seroprevalence of SARS-CoV-2 among the Health Care Workers in a Tertiary Care Centre, South India

**DOI:** 10.3390/vaccines10111967

**Published:** 2022-11-19

**Authors:** Divyaa Elangovan, Shifa Meharaj Shaik Hussain, Somasunder Virudhunagar Muthuprakash, Nanthini Devi Periadurai, Ashok Viswanath Nalankilli, Harshada Volvoikar, Preethy Ramani, Jayanthi Sivasubramaniam, Kalyani Mohanram, Krishna Mohan Surapaneni

**Affiliations:** 1Department of Microbiology, Panimalar Medical College Hospital & Research Institute, Varadharajapuram, Poonamallee, Chennai 600123, India; 2Department of Molecular Virology, Panimalar Medical College Hospital Research Institute, Varadharajapuram, Poonamallee, Chennai 600123, India; 3SMAART Population Health Informatics Intervention Center, Foundation of Healthcare Technologies Society, Panimalar Medical College Hospital & Research Institute, Varadharajapuram, Poonamallee, Chennai 600123, India; 4Departments of Biochemistry, Medical Education, Research, Clinical Skills & Simulation, Panimalar Medical College Hospital & Research Institute, Varadharajapuram, Poonamallee, Chennai 600123, India

**Keywords:** health care workers, safety, vaccine, COVID-19, IgG, ELISA

## Abstract

Global vaccine development efforts have been accelerated in response to the devastating COVID-19 pandemic. The study aims to determine the seroprevalence of SARS-CoV-2 IgG antibodies among vaccine-naïve healthcare workers and to describe the impact of vaccination roll-out on COVID-19 antibody prevalence among the health care centers in tertiary care centers in South India. Serum samples collected from vaccinated and unvaccinated health care workers between January 2021 and April 2021were subjected to COVID-19 IgG ELISA, and adverse effects after the first and second dose of receiving the Covishield vaccine were recorded. The vaccinated group was followed for a COVID-19 breakthrough infection for a period of 6 months. Among the recruited HCW, 156 and 157 participants were from the vaccinated and unvaccinated group, respectively. The seroprevalence (COVID-19 IgG ELISA) among the vaccinated and unvaccinated Health Care Workers (HCW) was 91.7% and 38.2%, respectively, which is statistically significant. Systemic and local side-effects after Covishield vaccination occur at lower frequencies than reported in phase 3 trials. Since the COVID-19 vaccine rollout has commenced in our tertiary care hospital, seropositivity for COVID-19 IgG has risen dramatically and clearly shows trends in vaccine-induced antibodies among the health care workers.

## 1. Introduction

COVID-19, a novel viral disease caused by SARS CoV-2 originated in Wuhan, China, during the investigation of a cluster of cases leading to unknown pneumonia in December 2019 [1,2]. SARS-CoV-2 rapidly spread worldwide and still poses a major challenge and threat to public health and healthcare systems [3]. Globally, during this study period, the WHO reported 364,191,494 confirmed cases of COVID-19, including 5,631,457 deaths [4,5]. Healthcare workers (HCW), including doctors, nurses and other paramedical staff, are the leading frontline personnel of a medical health care system. Due to the prolonged period of exposure, HCW are the most vulnerable cohort at a high risk of COVID-19 infections compared to the general population [6]. Infected HCW may pose risk to patients, family members and to the community as well. Therefore, the safety of HCW is essential to safeguard continuous patient care. WHO reports have documented that until 2020, at least 90,000 healthcare workers had been infected by COVID-19 [7].

Serological tests can provide more information on SARS CoV-2 infection as an antibody response of IgG formed following infection [8]. Antibody tests are helpful for detecting previous infection when measured two weeks after the onset of symptoms, but the duration of elevated antibody levels remains unknown. Studies on antibodies against SARS-CoV-2 are important, as it would decrease the number of virions that could infect ACE-2 receptor-expressing cells. The World Health Organization (WHO) has approved the performance of serosurveys in order to estimate the extent of COVID-19 infection in a population group and understand the disease dynamics of COVID-19 transmission [9].

India introduced a mass COVID-19 vaccination programme (Covishield and Covaxin) with two candidate vaccines from 16 January 2021 after the Emergency Use Approval [10]. In many countries, Health Care Workers (HCW) was among the first to be vaccinated. All the Health Care Workers (HCW) in our tertiary care hospital completed two doses of Covishield vaccine by May 2021. The Phase III data for Covishield from randomized clinical trials (RCTs) shows that the vaccine was safe and effective [11]. However, there is still a paucity of information as to the level of immune response this novel vaccine elicits, both at a humoral and cellular level in the community.

We undertook this study to determine the seroprevalence of IgG antibodies in SARS-CoV-2 among vaccine-naïve health care workers in our tertiary care hospital and to further describe the impact of vaccination roll-out on COVID-19 antibody prevalence among HCW. The study was also designed to track adverse effects of Covishield vaccine after the first and second dose of the vaccine. 

## 2. Material and Methods

This cross-sectional serosurvey was performed among the health care workers in a tertiary care hospital, in Tamil Nadu, South India. This serosurvey was conducted at two different time periods; first during January 2021, before the initiation of the COVID-19 vaccination to HCW by the Government of Tamil Nadu and second during April 2021, when all the health care workers in our hospital had been given the second dose of the COVID-19 vaccine. Institutional Human Ethics Committee of Panimalar Medical College Hospital & Research Institute (PMCHRI-IHEC) approval has been obtained prior to start of the study (Approval Number: PMCH&RI/IHEC/2020/029; dated: 30.12.2020). Following the approval and clearance from the Institutional Human Ethics Committee (PMCHRI-IHEC), eligible individuals were included in the study. The study was conducted according to the Declaration of Helsinki as the current study involves human subjects.

Individuals who agreed to participate answered an interview-based structured questionnaire after providing written informed consent. The questionnaire comprised questions relating to socio-demographic variables, including age, gender and respiratory symptoms or fever in the 6 months prior to enrolment in the study, hospitalization for COVID-19 since March 2020 and usage of masks in the workplace. In addition, the vaccinated health care worker recruited during April 2021 were asked to provide the adverse effects experienced within 48–72 hours and after 7 days of the first and second dose of the Covishield vaccination. Both systemic and local effects of the Covishield vaccination were taken into account. After informed consent, 5 mL serum sample were drawn from the participants and transported to the laboratory immediately, where they were centrifuged. Serum samples were stored at −70 °C until IgG ELISA testing was carried out, as illustrated the study flow diagram (Figure 1).

SARS-CoV-2 serological testing was performed using SARS-CoV-2 ELISA IgG assay (Euroimmun, Lübeck, Germany) in an automated analyzer targeting the S1 domain, including the receptor-binding domain that detects the presence of IgG antibodies against SARS-CoV-2 S proteins in human serum. Results are expressed as a ratio, calculated by dividing the optical densities of the sample by those of an internal calibrator provided with the test kit. The cut-off for samples to be considered positive was ≥ 1.1. The sensitivity and specificity of the SARS CoV-2 IgG ELISA kit was found to be 95% and 96.2%, respectively [12]. 

Subsequently, the vaccinated group was followed for a COVID-19 breakthrough infection for a period of 6 months. They were grouped into asymptomatic; symptomatic but not RT-PCR proven; and symptomatic, RT-PCR proven. We defined a breakthrough infection as a COVID-19 infection that was contracted on or after the 14th day of vaccination. 

## 3. Statistical Analysis

Data were analyzed using STATA 15.0 (Stata Corp, College Station, TX, USA). Values were expressed as a median, quartiles, frequency and percentages to understand the nature of the data. A chi-square test was used to assess the association between the vaccination status and the profile of the participants. A nonparametric Mann–Whitney U test was used to identify the significance of the ELISA values observed between the vaccinated and unvaccinated population, also used in other subgroup comparison analysis. An upset plot was used to present the occurrence of the symptoms after the 1st and 2nd dose of the COVID-19 vaccination. A smoothed density plot was presented to disseminate the observed ELISA value over the vaccinated and unvaccinated population.

## 4. Results

A total number of approximately 520 Health care workers were working in our tertiary care hospital during the study period (January to April 2021), of whom 313 HCW were willing to participate in the study. Among the HCW, 157 participants were initially unwilling to take the COVID-19 vaccination due to vaccine hesitancy. The first set of samples were collected from the unvaccinated HCW (*n* = 157) and remaining (*n* = 156) were taken after the second dose of the COVID-19 vaccination (IQR 12–14 days). The sample population comprised 107 males, of whom 48.6%and 51.4%were vaccinated and unvaccinated, respectively. Among the 206 female HCW, 50.5% and 49.5% were vaccinated and unvaccinated, respectively. In both vaccinated and unvaccinated individuals, the majority were in the age group between 20–35 years. In sub-population analysis, paramedical staff was represented in higher numbers among the vaccinated group; in contrast, nonmedical staff was a larger proportion in unvaccinated HCW, which is statistically significant (*P* < 0.0001). During the time of recruitment of HCW into the study, only ten percent of HCW had contracted a RT-PCR proven COVID-19 infection in the past 6 months. Surgical masks (58%) were the most common type of mask used for protection by the HCW, followed by the N95 mask (20%) and cloth mask (21%). The seroprevalence (COVID-19 IgG ELISA) among the vaccinated and unvaccinated HCW were 91.7% and 38.2%, respectively. As shown in Table 1, seropositivity for COVID-19 IgG ELISA was higher (70.4%) among vaccinated HCW than the vaccine-naïve HCW (29.6%), which is statistically significant (*P* < 0.0001).The majority (87.0%) of the seropositive individuals among the unvaccinated group did not report any symptoms related to COVID-19 infection at the time of the study nor in the past. Age, gender and the history of at least one self-reported symptom suggestive of COVID-19 in the last three months before the study were not associated with positive status (*P* > 0.05). 

We compared the COVID-19 IgG ELISA ratio between the vaccinated and unvaccinated HCW with the demographic and clinical characteristics of study population. There is a significant rise in COVID-19 IgG seropositivity in the vaccinated group among the age category of 20–35 years in comparison to the age group of above 35 years. There is no gender difference in COVID-19 IgG positivity both in the vaccinated and unvaccinated group. In the sub-population analysis, seropositivity was significantly higher in non-medical category compared to paramedical and medical faculties. The unvaccinated HCW who had a prior history of COVID-19 infection in the past 6 months showed the baseline COVID-19 IgG ratio of 3.26 (1.69–3.70). This was doubled in the vaccinated group with a COVID-19 IgG ratio of 6.69 (5.12–8.80). The seropositivity ratio of 5.27 (3.28–8.10) was significantly higher in the vaccinated HCW than the unvaccinated group with 0.36 (0.13–1.50) who had no positive history of COVID-19 infection and was found to be statistically significant. There is a stronger association of seropositivity with the usage of cloth and surgical mask than with N95 mask among the vaccinated group, as shown in Table 2.

Overall, side effects reported by vaccine recipients after the first dose was 68%, while 32% had side effects after the second dose of the Covishield vaccine. The most common symptoms after the first dose of COVID-19 vaccine were pain at the injection site (82%), body pain (33%) and low grade fever (30%), as shown in Figure 2 and Figure 3. The frequent combination of symptoms encountered were body pain with low grade fever (12%) and in combination with pain at injected site (12%). Two percent (2%) of patients did not elucidate any adverse effects after the first dose of the COVID-19 vaccine, whereas after the second dose of the COVID-19 vaccine, twenty percent (20%) had no adverse effects. Body pain and low grade fever were predominantly seen after the second dose of the COVID-19 vaccine. Certain side-effects, such as vomiting, syncope and allergy, were not observed in the vaccine recipients after first and second dose of COVID-19 vaccine.

There is a statistically significant (*P* < 0.001) increase in the COVID-19 IgG ELISA ratio between the vaccinated (5.87) and unvaccinated (2.71) HCW, as seen in Figure 4. Eight percent remained seronegative even after the two doses of the COVID-19 vaccine.

Among the subjects vaccinated with both doses of the vaccine, 2% were RT-PCR positive within 60 days after the second dose. They were admitted to the ward with mild symptoms and did not require oxygenation or critical care support. Although there were no RT-PCR positive cases during the further follow up of 180 days, 2–3% had at least one COVID-19 symptom, which was not confirmed by molecular testing. The difference in COVID-19 IgG antibody levels among the asymptomatic, symptomatic RT-PCR proven and unproven cases in vaccinated individuals during the 6 months follow up has been depicted in the Figure 5.

## 5. Discussion

The seroprevalence status of SARS-Cov-2 among the vaccine-naïve HCW in our centre was 38% in January 2021, which is almost one year after the index COVID-19 case was identified in India. Seroprevalence among HCW (38%) observed in this study was a little higher compared with the 26% prevalence estimated in a large sero-surveillance conducted among the HCW during the same period between December 2020 to January 2021 in India [13,14]. The confounding factors, such as the age and sex of the unvaccinated group, did not show much difference in COVID-19 IgG seropositivity. We identified variations in the seroprevalence of SARS-CoV-2 antibodies among the different groups of healthcare workers. 

The highest seroprevalence was observed in the non-medical (20%) category with a lower seroprevalence among medical (9%) and para-medical (9%) HCW. This could be due to the differential risks of SARS-CoV-2 exposure that exist within the hospital environment and strict adherence to PPE by medical and paramedical workers. Furthermore, the study demonstrates that the magnitude of COVID-19 antibody responses were significantly greater in individuals with prior symptomatic illness compared with those who remained asymptomatic. This result was found to be consistent with a similar report from a hospital in northern India [15].

The rates of side-effects following the Covishield vaccine were lower than expected. The phase 2–3 trial of the ChAdOx1 nCoV-19 vaccine reported local and systemic adverse effects in 88% and 72% of participants who received the first injection and second dose, respectively [16,17]. In contrast, we found a lower rate of adverse effects of 68% after the first dose and 32% after the second dose of Covishield. None of the participants included in this report had any suspected unexpected serious adverse reactions, as observed in phase 2–3trials [18].

This cross-sectional study reported an overall 91.7% (143/156) seropositivity rate after two complete doses of vaccines in all the study participants. Similarly high seropositivity rate of (95%) were reported in the cross-sectional study of corona virus vaccine-induced antibody titre (COVAT) conducted in West Bengal [19]. Although there was no significant rise in COVID-19 IgG ratio amongst age and gender variables, the nonmedical category demonstrated increased COVID-19 IgG value in comparison to other categories. Various multicentric studies highlight the robust immune response that was produced by a natural COVID-19 infection and the additional protective effect that vaccination contributed to it [20,21]. Similarly, we observed the considerable increase in COVID-19 vaccine-induced antibody levels in those vaccinated HCW who had a history of COVID-19 in the past 6 months.

Eight percent (8%) of healthy HCW did not elucidate any immune response to vaccination against SARS-CoV-2. It is unclear whether this may lead to reduced protection from COVID-19 infection and disease [22,23,24]. Further immunological studies were needed to confirm the long-term effectiveness of SARS-CoV-2 vaccines and to determine the duration of protection in order to assess the need and ideal schedule for revaccination.

The difference in COVID-19 breakthrough infection has been more noticeable in the period after the Delta variant became dominant [25,26]. Overall, 2% of patients had a breakthrough infection during the follow up after completing the course of vaccination. Although the emergence of the Delta variant in India was devastating with high mortality, breakthrough cases tended to be substantially less severe compared with pre-vaccination COVID-19 cases, regardless of a person’s immune status. The data confirmed that SARS-CoV-2 vaccinations are highly successful and the importance of full vaccination for preventing breakthrough infection is emphasized [27].

To the best of our knowledge, this cross-sectional sero-surveillance study is the first of its kind that has involved HCW from Southern India reporting anti-spike antibody kinetics among the unvaccinated and vaccinated populations. However, we also acknowledge several limitations in the present study; first, we screened the two different cohorts for COVID-19 IgG sero-surveillance because of the logistic issue. Ideally, a baseline COVID-19 IgG titre along with two values of anti-spike antibody after the first and second dose of the Covishield vaccine would have added more value in inferring the immune response to COVID-19 vaccine. Second, we measured only anti-spike binding antibody and could not assess NAb and cell-mediated immune responses [28,29,30], although a recent study has demonstrated a high correlation between spike protein-based ELISA and different antibody classes, including NAb in COVID-19 patients.

## 6. Conclusions

Systemic and local side-effects after Covishield vaccination occurred at lower frequencies than reported in phase 3 trials. Since the COVID-19 vaccine rollout commenced in our tertiary care hospital, seropositivity for COVID-19 IgG has risen dramatically and clearly shows trends in vaccine-induced antibodies among the health care workers. This adds to the evidence for the impact of the COVID-19 vaccine on the seroprevalence of SARS-CoV-2. Future studies to identify the protective thresholds of antibody responses may help in triaging the HCW who are at greatest risk for breakthrough infections.

## Figures and Tables

**Figure 1 vaccines-10-01967-f001:**
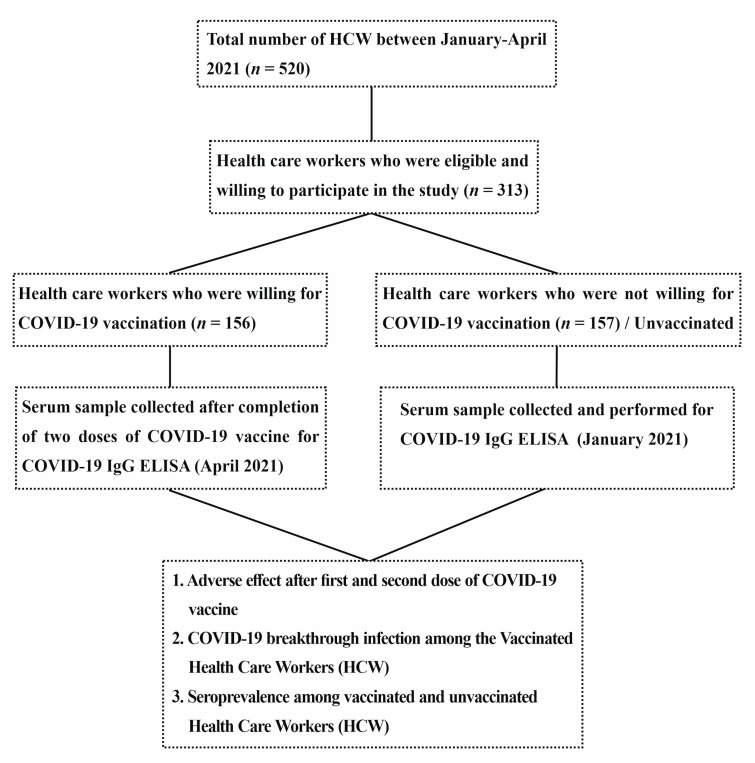
Study flow diagram.

**Figure 2 vaccines-10-01967-f002:**
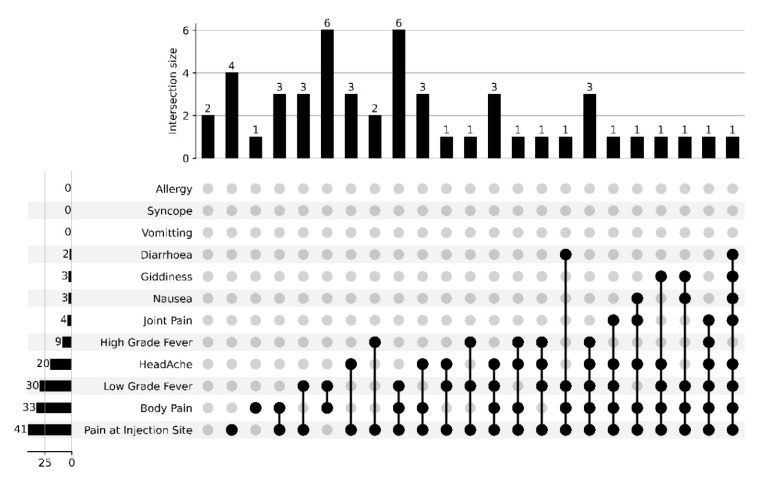
Upset plot representing reported side-effects and co-existence of symptoms after the first dose of the vaccine. The values represent the number of individuals experiencing a symptom category or combination of categories. Black lines link multiple symptoms indicated by black dots.

**Figure 3 vaccines-10-01967-f003:**
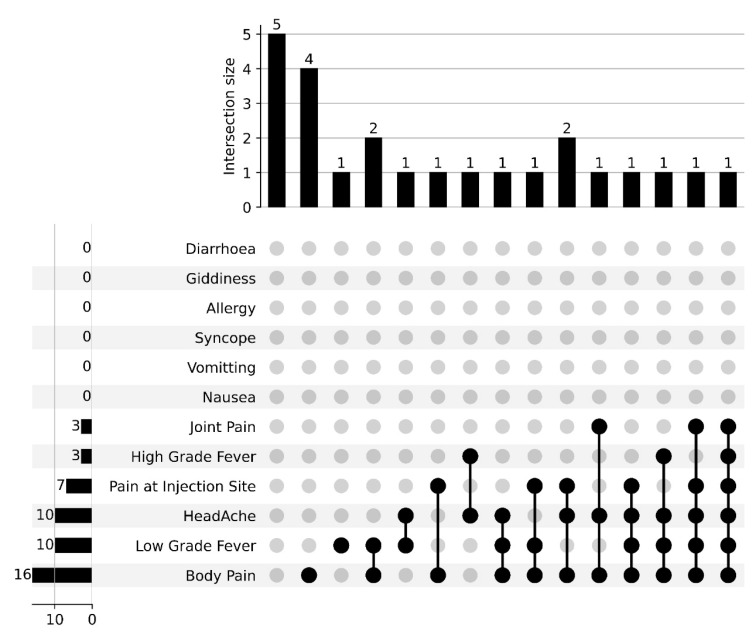
Upset plot representing reported side-effects and co-existence of symptoms after the second dose of the vaccine. The values represent the number of individuals experiencing a symptom category or combination of categories. Black lines link multiple symptoms indicated by black dots.

**Figure 4 vaccines-10-01967-f004:**
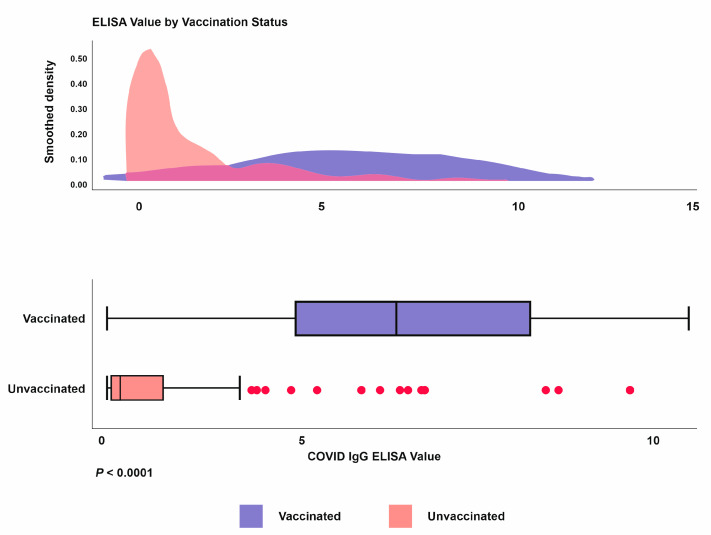
Density distribution and comparison of COVID-19 IgG ELISA value among vaccinated and non-vaccinated individuals. The box plots show the distribution of ELISA between the vaccinated and non-vaccinated groups (*P* < 0.0001 shows the significant difference using the Mann–Whitney U test).

**Figure 5 vaccines-10-01967-f005:**
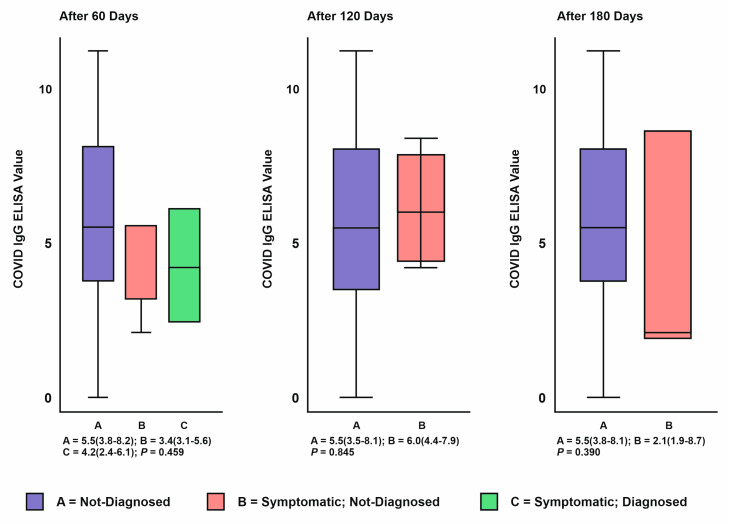
Difference in COVID-19 IgG antibody levels among the symptomatic RT-PCR proven and unproven cases in vaccinated individuals during the 6 months follow up.

**Table 1 vaccines-10-01967-t001:** Characteristics of study participants by their vaccination status.

Factors	Vaccinated	Unvaccinated	*p* Value ^a^
	156 (49.8%)	157 (50.2%)	
Age Group			
20–25	47 (43.5%)	61 (56.5%)	0.4250
26–35	63 (52.5%)	57 (47.5%)	
36–55	35 (55.6%)	28 (44.4%)	
>55	11 (50.0%)	11 (50.0%)	
Gender			
Male	52 (48.6%)	55 (51.4%)	0.7510
Female	104 (50.5%)	102 (49.5%)	
Category			
Non-Medical	29 (28.2%)	74 (71.8%)	<0.0001
Para-Medical	80 (63.5%)	46 (36.5%)	
Medical	47 (56.0%)	37 (44.0%)	
Past History of COVID-19 Infection			
Yes	22 (71.0%)	9 (29.0%)	0.0130
No	134 (47.5%)	148 (52.5%)	
Type of Masks used for Protection			
Cloth	8 (11.9)	59 (88.1%)	<0.0001
Surgical	96 (52.5)	87 (47.5%)	
N95	52 (82.5)	11 (17.5%)	
ELISA Test			
Negative (<1.1)	13 (11.8%)	97 (88.2%)	<0.0001
Positive (≥1.1)	143 (70.4%)	60 (29.6%)	

^a^ Chi-Square test was used to assess the association between the factors and vaccination status.

**Table 2 vaccines-10-01967-t002:** Association of demographic and clinical characteristics of vaccinated and unvaccinated individuals with a COVID-19 IgG ELISA profile.

Factors	Vaccinated		*p* Value ^b^
	Yes ^a^	No ^a^	
Age Group			
20–25	5.74 (3.83–7.60)	0.43 (0.14–1.50)	<0.0001
26–35	5.48 (3.49–8.00)	0.28 (0.14–1.50)	<0.0001
36–55	5.33 (2.23–8.10)	0.54 (0.12–3.70)	0.0001
>55	5.50 (3.77–8.40)	1.50 (0.12–3.70)	0.0326
Gender			
Male	5.26 (3.82–7.90)	0.33 (0.10–1.30)	<0.0001
Female	5.52 (3.39–8.20)	0.48 (0.15–2.10)	<0.0001
Category			
Non-Medical	7.44 (4.94–8.60)	0.50 (0.15–2.00)	<0.0001
Para-Medical	5.35 (3.36–7.50)	0.18 (0.15–1.50)	<0.0001
Medical	5.18 (3.25–8.10)	0.59 (0.10–2.10)	<0.0001
History of past COVID-19 Infection			
Yes	6.69 (5.12–8.80)	3.26 (1.69–3.70)	0.0014
No	5.27 (3.28–8.10)	0.36 (0.13–1.50)	<0.0001
Type of Masks used for Protection			
Cloth	6.35 (3.24–9.00)	0.43 (0.10–1.50)	<0.0001
Surgical	5.68 (3.26–7.90)	0.42 (0.14–1.70)	<0.0001
N95	5.04 (3.79–8.10)	0.55 (0.13–3.70)	0.0013
ELISA Test			
Negative (<1.1)	0.07 (0.05–0.30)	0.15 (0.10–0.30)	0.0379
Positive (≥1.1)	5.87 (4.39–8.40)	2.71 (1.50–4.20)	<0.0001

^a^ Values were shown as median (first quartile–third quartile). ^b^ Mann–Whitney test was used to compare the ELISA Value between the Vaccinated vs. Unvaccinated.

## Data Availability

Data will be made available on request.

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
