# Peer review of "Impact of COVID-19 Vaccination on Seroprevalence of SARS-CoV-2 among the Health Care Workers in a Tertiary Care Centre, South India"

_vaccines, 2022, doi:10.3390/vaccines10111967_

Round 1
Reviewer 1 Report
The study presented aims to study a group of healthcare professionals with the objective of calculating the prevalence of COVID infection prior to vaccination, assessing the impact of vaccination and describing the adverse effects of the vaccines administered. The study is interesting and I congratulate the authors for their work.
After reading it thoroughly I have some comments to make that I hope will help the authors to improve it.
1. In the introduction I would suggest making minor changes. In line 51 I suggest to remove "severe acute...." and leave it with "SARS-COV2 virus". Line 53 and 54 I suggest to put it after line 55 and 56.
2. Methods:
- Better define the study periods.
- Explain what will be done in each study period.
- Define and describe what the study population is like in each period (is it the same, how is it recruited,...).
- When the questionnaire is passed to them
- how to collect adverse effects
- Describe the losses and the reason
- Rethink the algorithm because it is not understood.
3. Results/Discussion:
- The prevalence found in the first group is not comparable to the second because they are different populations and at different times.
- We cannot know the impact of vaccination because the incidence of virus infection has not been systematically studied or is not presented in the results.
- It is not clear how adverse effects are recorded.
- Is the subgroup of vaccinated patients followed up? It is not explained in the methods.
I believe that the study has many limitations that should be explained in the discussion to know or intuit how they can influence the results and conclusions.
In summary, it would have been better to calculate the prevalence of antibodies in all the health workers recruited (313) and then to carry out two subgroups (vaccinated/non-vaccinated) and to study antibodies in each of them again. This would allow them to compare and estimate the impact of vaccination, controlling some of the many biases that can influence these types of studies.
In my opinion the article requires a major revision and a rethinking of the design and the objectives
Author Response
|
Comment No |
Comment |
Response |
|
1 |
In the introduction I would suggest making minor changes. In line 51 I suggest to remove "severe acute...." and leave it with "SARS-COV2 virus". Line 53 and 54 I suggest to put it after line 55 and 56. |
Changed as per the Reviewer’s suggestion |
|
2 |
Methods: - Better define the study periods. - Explain what will be done in each study period. - Define and describe what the study population is like in each period (is it the same, how is it recruited,...). - When the questionnaire is passed to them - how to collect adverse effects - Describe the losses and the reason - Rethink the algorithm because it is not understood.
|
Addressed in revised manuscript |
|
3 |
Results/Discussion: - The prevalence found in the first group is not comparable to the second because they are different populations and at different times. - We cannot know the impact of vaccination because the incidence of virus infection has not been systematically studied or is not presented in the results. - It is not clear how adverse effects are recorded. - Is the subgroup of vaccinated patients followed up? It is not explained in the methods.
I believe that the study has many limitations that should be explained in the discussion to know or intuit how they can influence the results and conclusions.
|
we acknowledge several limitation in the present study, first we have screened the two different cohorts for COVID IgG serosurveillance because of the logistic issue. Ideally, baseline COVID IgG titre along with two values of anti-spike antibody after first and second dose of covishield vaccine would have added more value in inferring the immune response to COVID vaccine. Yes, We cannot know the impact of vaccination because the incidence of virus infection has not been systematically studied , but we extrapolate the seroprevalence of vaccinated individuals would also be in lower side as seen in unvaccinated HCW |
|
4 |
In summary, it would have been better to calculate the prevalence of antibodies in all the health workers recruited (313) and then to carry out two subgroups (vaccinated/non-vaccinated) and to study antibodies in each of them again. This would allow them to compare and estimate the impact of vaccination, controlling some of the many biases that can influence these types of studies. |
We have done the subgroup analysis as depicted in fig 4 and fig 5 |

Reviewer 2 Report
In this manuscript, Mohan and co-workers reported their statistical results of a sero-survey performed in a tertiary care center, Tamil Nadu in South India. The aim is to compare the level of COVID IgG antibodies between vaccinated and unvaccinated health care workers (HCW) and to realize the impact of Covishield vaccination on sero-prevalence of SARS-CoV-2. The participants involved 157 vaccinated HCW and 156 unvaccinated HCW from the tertiary care center. For unvaccinated individuals, their serum samples were collected in January 2021, which was at the end of COVID wave 1 before initiation of COVID vaccination to HCW by the government. For vaccinated ones, their samples were collected in April 2021 after they received two doses of Covishield vaccine. In addition, the adverse effects of Covishield vaccine were traced after the participants received the first and second dose of vaccine along with the occurrence of COVID breakthrough infection for a period of 6 months.
According to the ELISA test results, authors found that the sero-positivity ratio of vaccinated participants was significantly higher than that of unvaccinated ones. They concluded that COVID vaccine clearly induced IgG antibody among the HCW. Additionally, Covishield vaccine caused systemic and local side-effects at lower frequencies than reported in phase 3 trials. The main side-effects were pain at the injection site, body pain, and low grade fever. The findings of this survey are informative and valuable for the execution of vaccination programs in the future. However, there are several places and points which need to be improved and corrected as described below:
1. The sample numbers of vaccinated and unvaccinated are inconsistent throughout the entire manuscript. They are 157 vaccinated and 156 unvaccinated (please see Abstract on page 1 and Figure 1 on page 3), but 156 vaccinated and 157 unvaccinated (please see the text in lines 131-133 and Table 1 on page 4). These numbrers have to be-rechecked and corrected.
2. Page 3, Figure 1: N-313 should be revised to N = 313; n-157 should be revised to N = 157 (or 156?); the line of box for “Serum sample collected after completion….” needs to be re-adjusted to show all of the words. Additionally, it is suggested to add “4. COVID infection for unvaccinated after their serum samples were collected” for comparison with COVID breakthrough infection.
3. The quality of Table 1 needs to be improved extensively. The numbers in the parenthesis are confusing and may cause some misleading. They are ratio between vaccinated and unvaccinated. Thus (43.5) should be (43.5%) and (56.5) should be (56.5%), etc. and thus revision of all of those related numbers is required. “Category” is not specific and needs to be revised. “Yes” and “No” have to be aligned vertically or centered. “ELISA Test Positive” is suggested to change to “ELISA Test”. The footnote, such as a and b, should be added to the last line for “Row percentage; Chi-Square test….” in the Table.
4. Page 4, line 144, 92% should be 91.7% as there are three significant figures for unvaccinated 38.2% and others reported in the manuscript.
5. Figures 1-4 and Tables 1 and 2 should be cited in the text. Moreover, their description should be presented before their related figures and tables.
6. Page 5, Table 2. “Vaccinated” should be centered between Yes and No. “Category” and “ELISA Test Positive” are suggested to revise as mentioned for Table 1. Again the footnote, such as a and b, should be added to the last line for “Values were….; Mann-Whitney test….” in the Table.
7. Page 6, Line 167, authors may have to check whether it is “three fold increase” …. between the vaccinated (5.87) and unvaccinated (2.71) HCW?
8. The resolution of words in Figures 2 and 3 is poor and has to be improved.
In addition to COVID breakthrough infection of vaccinated individuals, it is equally important to trace the infection status of the unvaccinated after their serum samples were collected. The factors worthy to investigate include severity of symptoms, hospitalization, mortality, etc. to compare with the vaccinated ones.
Overall, this manuscript needs major revision. It is not recommended for publication in Vaccines at its current form.
Author Response
|
Comment No |
Comment |
Response |
|
1 |
The sample numbers of vaccinated and unvaccinated are inconsistent throughout the entire manuscript. They are 157 vaccinated and 156 unvaccinated (please see Abstract on page 1 and Figure 1 on page 3), but 156 vaccinated and 157 unvaccinated (please see the text in lines 131-133 and Table 1 on page 4). These numbrers have to be-rechecked and corrected. |
Sample numbers were rechecked and corrected throughout the manuscript |
|
2 |
Page 3, Figure 1: N-313 should be revised to N = 313; n-157 should be revised to N = 157 (or 156?); the line of box for “Serum sample collected after completion….” needs to be re-adjusted to show all of the words. Additionally, it is suggested to add “4. COVID infection for unvaccinated after their serum samples were collected” for comparison with COVID breakthrough infection. |
Changes made accordingly |
|
3 |
The quality of Table 1 needs to be improved extensively. The numbers in the parenthesis are confusing and may cause some misleading. They are ratio between vaccinated and unvaccinated. Thus (43.5) should be (43.5%) and (56.5) should be (56.5%), etc. and thus revision of all of those related numbers is required. “Category” is not specific and needs to be revised. “Yes” and “No” have to be aligned vertically or centered. “ELISA Test Positive” is suggested to change to “ELISA Test”. The footnote, such as a and b, should be added to the last line for “Row percentage; Chi-Square test….” in the Table. |
Changes made accordingly |
|
4 |
Page 4, line 144, 92% should be 91.7% as there are three significant figures for unvaccinated 38.2% and others reported in the manuscript. |
Changes made accordingly |
|
5 |
Figures 1-4 and Tables 1 and 2 should be cited in the text. Moreover, their description should be presented before their related figures and tables. |
Figures and tables cited in the text |
|
6 |
Page 5, Table 2. “Vaccinated” should be centered between Yes and No. “Category” and “ELISA Test Positive” are suggested to revise as mentioned for Table 1. Again the footnote, such as a and b, should be added to the last line for “Values were….; Mann-Whitney test….” in the Table. |
Changes made accordingly |
|
7 |
Page 6, Line 167, authors may have to check whether it is “three fold increase” …. between the vaccinated (5.87) and unvaccinated (2.71) HCW? |
Yes, there is significant increase in COVID IgG ELISA ratio |
|
8 |
The resolution of words in Figures 2 and 3 is poor and has to be improved.
In addition to COVID breakthrough infection of vaccinated individuals, it is equally important to trace the infection status of the unvaccinated after their serum samples were collected. The factors worthy to investigate include severity of symptoms, hospitalization, mortality, etc. to compare with the vaccinated ones. |
Resolution of images has been changed according to the journal requirements.We could not follow up unvaccinated individuals for COVID breakthrough infections,as they were also vaccinated over the time period during follow up. |

Round 2
Reviewer 2 Report
Though the authors have responded to all of the questions and made some revisions and corrections on the manuscript, its quality is still not improved enough to meet the standard of its publication in Vaccines. It seems that the revision was done effortlessly. For example, Figure 1 is lack of aesthetics. The arrows were drawn randomly and should be aligned, such as centered and symmetrically. Furthermore, the space between words should be consistent.
Figures 2 and 3 are upset plots and were studied for the same type of subjects. The sizes of their words in both figures and the scale of bars should be consistent too. Those art work represents the quality of the manuscript and should be conducted by authors before their manuscript is accepted.
Moreover, Figure 1 is still not cited in the main text. The point of citation of Table 1 in the main text cannot reflect those mentioned numbers in Table 1 (i.e., 91.7 % and 38.2%). On page 7, line 180, figure 2 should be changed to Figure 2 to make it consistent with Figure 3.
Overall, this manuscript needs extensive editing work.
Author Response
Response to Reviewer -2 Comments
Point 1: Figure 1 is lack of aesthetics. The arrows were drawn randomly and should be aligned, such as centered and symmetrically. Furthermore, the space between words should be consistent.
Response 1: Figure 1 has been corrected accordingly as per reviewer comments
Point 2: Figures 2 and 3 are upset plots and were studied for the same type of subjects. The sizes of their words in both figures and the scale of bars should be consistent too.
Response 2: Yes, upset plot has been modified as per reviewer suggestions and they were studied in the same population , but the the number of individuals experiencing complex of symptoms were more after the first dose of Covid Vaccine than second dose
Point 3: Figure 1 is still not cited in the main text.
Response 3: Figure 1 is cited in the text as per reviewer comments
Point 4: The point of citation of Table 1 in the main text cannot reflect those mentioned numbers in Table 1 (i.e., 91.7 % and 38.2%).
Response 4: This has been addressed in the text and changed as follows -As shown in Table 1, seropositivity for COVID IgG ELISA was higher (70.4%) among the vaccinated HCW than the vaccine naive HCW (29.6%) which is statistically significant(p<0.0001).
Point 5: On page 7, line 180, figure 2 should be changed to Figure 2 to make it consistent with Figure 3.
Response 5: Correction has been done in the text
